# VEGF Upregulates EGFR Expression to Stimulate Chemotactic Behaviors in the rMC-1 Model of Müller Glia

**DOI:** 10.3390/brainsci10060330

**Published:** 2020-05-29

**Authors:** Juan S. Peña, Maribel Vazquez

**Affiliations:** Department of Biomedical Engineering, Rutgers, The State University of New Jersey, Piscataway, NJ 08854, USA; juan.s.pena@rutgers.edu

**Keywords:** Müller glia, microfluidics, VEGF, EGF-R, reactive gliosis, migration assays, blood retinal barrier

## Abstract

Progressive vision loss in adults has become increasingly prevalent worldwide due to retinopathies associated with aging, genetics, and epigenetic factors that damage the retinal microvasculature. Insufficient supply of oxygen and/or nutrients upregulates factors such as vascular endothelial growth factor (VEGF) and epidermal growth factor (EGF), which can induce abnormal angiogenesis and damage the structural arrangement of the retinal blood barrier (BRB). Müller glia (MG) regulate the diffusion of essential compounds across the BRB and respond to retinal insults via reactive gliosis, which includes cell hypertrophy, migration, and/or proliferation near areas of elevated VEGF concentration. Increasing concentrations of exogenous VEGF, upregulated by retinal pigmented epithelium cells, and endogenous epidermal growth factor receptor (EGF-R) stimulation in MG, implicated in MG proliferative and migratory behavior, often lead to progressive and permanent vision loss. Our project examined the chemotactic responses of the rMC-1 cell line, a mammalian MG model, toward VEGF and EGF signaling fields in transwell assays, and within respective concentration gradient fields produced in the glia line (gLL) microfluidic system previously described by our group. rMC-1 receptor expression in defined ligand fields was also evaluated using quantitative polymerase chain reaction (qPCR) and immunocytochemical staining. Results illustrate dramatic increases in rMC-1 chemotactic responses towards EGF gradient fields after pre-treatment with VEGF. In addition, qPCR illustrated significant upregulation of EGF-R upon VEGF pre-treatment, which was higher than that induced by its cognate ligand, EGF. These results suggest interplay of molecular pathways between VEGF and EGF-R that have remained understudied in MG but are significant to the development of effective anti-VEGF treatments needed for a variety of retinopathies.

## 1. Introduction

Increasing numbers of adults worldwide are experiencing progressive and permanent loss of vision [1,2]. Many visual impairments are a result of retinal degeneration from inherited and age-related diseases, as well as from chronic health conditions such as cardiovascular disease and diabetes [3]. Müller glia (MG) are central to the ocular response to retinal insult and have become recent therapeutic targets for diabetic retinopathy, glaucoma, and age-related macular degeneration [4,5]. MG of the visual system provide structural support for neurosensory retina, preserve the homeostasis of retinal neurons critical to phototransduction [6,7], and regulate the diffusion of essential compounds across the blood retinal barrier (BRB). As shown in Figure 1, MG cellular processes form the basement membrane of capillaries within the retina [8] and control transport of oxygen and nutrients from vascular networks [9].

Breakdown of the BRB and occlusion of microcapillaries often leads to progressive retinal dysfunction, characterized by ischemia and/or hypoxia, i.e., diminished and/or insufficient levels of blood and oxygen in retinal tissue, respectively [11,12]. In response, MG upregulate production of pro-angiogenic factors, such as vascular endothelial growth factor (VEGF) and fibroblast growth factor (FGF) [13], to increase blood circulation and support the elevated metabolic activities of damaged retina. Temporal expression of endogenous VEGF promotes axonal growth, post-ischemia, and aids reorganization of glial processes within retinal tissue [14,15,16,17]. However, the nature and duration of VEGF stimulus predicates MG neuroprotective or neurodegenerative behavior [18,19,20]. Unrestricted VEGF upregulation leads to BRB breakdown, neovascularization, and vascular leakage [21], whereas chronically depressed VEGF can reduce the retinal density of MG to stimulate degeneration [22]. The impacts of altered VEGF signaling in retinal tissue has become critical to contemporary therapy, as recent clinical trials of anti-VEGF compounds, e.g., Lucentis and Avastin [23,24,25,26], have recorded retinal thinning and decreased MG structural support in patients after long-term treatment [21,22].

VEGF expression impacts multiple signaling pathways central to homeostasis and repair through cross-talk with Akt/PBK and MEK [26]. The relationship(s) between VEGF and signaling through the epidermal growth factor receptor (EGF-R) have been well studied in the nervous system, where VEGF stimulates de/differentiation of neural stem-like cells and upregulation of EGF-R for essential migratory responses during development, tissue repair, and oncogenesis [27,28,29]. In the retina, VEGF paracrine signaling has been correlated with upregulation of the Heparin Binding EGF ligand (HB-EGF) after injury, which acts through EGF-R [30]. Upregulation of EGF-R via HB-EGF stimulates MG to exit quiescent states and initiate gliosis responses for tissue repair [30,31]. Recent work from our group have used the rMC-1 cell line as a model of MG in reactive state [32,33]. Specifically, rMC-1 express glial fibrillary acid protein (GFAP), a marker for reactive gliosis, cellular retinaldehyde-binding protein (CRALBP), a highly-specific MG marker in adult retina, and glutamine transporters, which are ubiquitous to MG. Additionally, our work has also elucidated EGFR-mediated chemotaxis in rMC-1 upon concentration gradients of EGF and VEGF [34]. However, the relationship between EGF-R expression and exogenous VEGF remains incompletely studied in retinal MG and may have significant impacts in emerging retinal therapies.

The current project examined the relationship between VEGF signaling and expression of EGF-R leading to rMC-1 chemotaxis, or directional migration, central to both neuroprotective and neurodisruptive glial responses [35]. Controlled signaling fields of VEGF and EGF were applied, separately, to stimulate the migration of rMC-1 within conventional transwell assays and within our microfluidic system, the glia line (gLL), previously developed by our group on the scale of adult retina to study MG within confined environments [33]. Results illustrated that both VEGF and EGF ligands stimulated rMC-1 chemotaxis with differences in migration distances and directionality of motion. However, our project recorded increases in rMC-1 cell trajectories in response to EGF signaling gradients after previous exposure to VEGF stimulus. Further, exogenous VEGF produced a near 20-fold increase in the EGF-R expression of rMC-1, but only an approximate 3-fold increase in the expression of its cognate receptor, VEGF-R. These results highlight unexamined crosstalk between VEGF and EGF-R signaling in rMC-1. Our data are among the first to examine these relationships in retinal rMC-1 and are significant to both current retinal treatments and emerging therapies targeting MG.

## 2. Materials and Methods

### 2.1. Cell Culture

MG in this project were modeled by cultured rMC-1 cells (Kerafast, Cat. No. ENW001), derived from rat primary MG cells and immortalized using SV40 T-antigen. These cells were selected because they express well-known markers characteristic of primary MG [36,37,38,39]. rMC-1 were cultured in Dulbecco’s Modified Eagle’s medium (DMEM) (Cat No. 30-2002, ATCC, VA) containing 4 mM L-glutamine, 4500 mg/L glucose, 1 mM sodium pyruvate, and 1500 mg/L sodium bicarbonate supplemented with 10% Fetal Bovine Serum (FBS) (Invitrogen-Gibco, Rockville, MD) at 37 °C and 5% CO_2_. Two weeks prior to experiments, cells were cultured in serum-restricted media (1% FBS in DMEM) at a concentration of 1.0 × 10^6^ cells/mL in T-75 flasks and passaged as needed. Cultured rMC-1 were dislodged using Accutase^®^ Solution (Cat No. 10210-214, VWR, PA) and re-suspended in 10% FBS supplemented DMEM for testing.

### 2.2. Gene Expression via Reverse Transcription Quantitative Polymerase Chain Reaction (qPCR)

Expression levels of EGF-R (epidermal growth factor receptor), VEGF-R (vascular endothelial growth factor receptor), FGFR-2 (fibroblast growth factor 2 receptor), and FGFR-8 (fibroblast growth factor 8 receptor) were measured using qPCR, and normalized to glyceraldehyde 3-phosphate dehydrogenase (GAPDH) gene expression, as previously described by our group [32]. rMC-1 cells were stimulated with ligands for 2 h prior to analyses. RNA was isolated and evaluated using Trizol protocol (Sigma-Aldrich, St. Louis, USA) and its concentration was measured photometrically in triplicate.

Automated PCR was performed in a final volume of 20 μL containing 5 ng of RNA template, 10 μL of TaqMan RT-PCR Mix, 1 μL of the Taqman Gene Expression Assay primer and 0.5 μL of Taqman RT Enzyme Mix in a StepOne Plus Real-Time PCR system (Applied Biosystems, USA, Waltham). Reverse transcription was performed at 48 °C for 15 min, followed by denaturation of the cDNA and activation of the DNA polymerase at 95 °C for 10 min. A total of 40 cycles of 15 sec at 95 °C and 1 min at the annealing temperature of 60 °C were used to amplify the PCR product. The relative change in expression levels between the control and experimental conditions was calculated using the conventional ΔΔCT method, as used previously by our group [32,40,41]. Primer sequences are provided in Table 1. The qPCR test was carried out prior to the migration assays yet using cells of the same batch.

### 2.3. Immunocytochemical Staining (ICC)

rMC-1 were plated at a concentration of 2.5 × 10^5^ cells/mL, and approximately 0.5 mL of cell solution was added to each borosilicate glass chamber (Cat No. 155383, ThermoFisher Scientific, MA) with serum-free media, and left to adhere for 24 h. Media was removed and the samples were washed 3 times with Dulbecco’s phosphate-buffered saline (DPBS; Cat No. D8537, Sigma-Aldrich, GA) solution, fixed with 10% formalin (Cat No. HT501128, Sigma-Aldrich, GA) for 10 min then rinsed twice with PBS. The samples were permeabilized with 0.1% Triton X (Cat No. 9002-93-1, Sigma-Aldrich, GA) in 0.1% Bovine Serum Albumin (BSA, Cat No. A7906, Sigma-Aldrich, GA) in DPBS for 10 min and then blocked with 1.0% BSA in PBS for 1 h. The samples were rinsed twice with the same blocking solution prior to being exposed to a 1:200 dilution of either EGF-R antibody, rabbit monoclonal (Cat No. 700308, ThermoFisher Scientific, IL) or VEGF receptor 2 monoclonal antibody (Cat No. MA5-15157, ThermoFisher Scientific, IL) in antibody diluent (Cat No. S3022, Agilent, CA) overnight at room temperature (25 °C). The samples were washed 3 times with DPBS, exposed to a 1:200 dilution of goat anti-rabbit igG (H+L) secondary antibody (Cat No. A32732, ThermoFisher Scientific, IL) for 30 min, and then rinsed 3 times with DPBS. Nuclear staining (Cat No. R37606, ThermoFisher, IL) was performed for 20 min at room temperature (25 °C), then samples were washed again with DPBS and covered with glycerol (Cat No. 15514011, ThermoFisher, IL) to prevent drying. All data were measured and compared to basal conditions in triplicate.

### 2.4. Transwell Assays

Boyden chambers (VWR, PA) in combination with thin porous membranes were used as transwell assays for this study. rMC-1 were seeded at a concentration of 2.5 × 10^5^ cells/mL in serum-free DMEM on the top part of a two compartment transwell model, divided by an 8 µm porous polyester membrane of 10µm thickness (Cat No. 3464, Corning Inc., ME), as per Figure 2. The bottom compartment was filled with either EGF or VEGF at 100ng/mL in DMEM for test conditions or serum-free DMEM for the control group. rMC-1 were allowed to migrate for 6 h at 37 °C and 5% CO_2_. Upon completion, the top surface of the membrane was cleaned with an applicator followed by fluorescent staining of the bottom surface of the membrane, containing the cells that migrated through it, using Cell TrackerTM Green CMFDA (Cat No. C7025, ThermoFisher, MA) at a concentration of 15 µM. Numbers of motile rMC-1 located at the bottom side of the porous membrane were optically measured for all tests. rMC-1 migration in response to ligand signaling through these transwell assays was examined using four conditions: (i) rMC-1 stimulated by EGF signaling fields; (ii) rMC-1 stimulated by VEGF signaling fields; (iii) rMC-1 exposed to VEGF for one hour and then stimulated by EGF signaling; and (iv) control (i.e., media only, no gradients). Each testing condition was examined in triplicate.

### 2.5. Overview of the gLL Microfluidic System

The transport phenomena governing the development of concentration gradients in the glial line system (gLL), has been previously described by our group [33], and used to examine the migration of glia derived from the central and peripheral nervous system to chemotactic and electrotactic stimuli. In brief, the gLL device consists of two volumetric reservoirs, a source and a sink, of 98 µL volume each, connected by an adjoining microchannel 1.3 cm in length (ℓc) and 192.6 µm in hydraulic diameter (D_H_), as per Figure 2. The device is cast in commercial polydimethylsiloxane (PDMS; Cat No. 1020992-312, VWR, PA) and bonded to a chemically cleaned microscope slide (Nanostrip, VWR, MA) using ozone treatment. Reagents inserted into the source reservoir transport along the microchannel to the opposite reservoir via the well-established convective-diffusion model shown below:(1)$$\frac{\partial C}{\partial t}+ū\cdot \frac{\partial C}{\partial x}=D\frac{\partial^{2}C}{\partial x^{2}}$$
where C (kg/m^3^)is the solution concentration, t (seconds) denotes time, ū (m/s) is bulk velocity, D (m^2^/s) is the diffusion coefficient, and x denotes position within the microchannel length [40].

Inner surfaces of multiple gLL devices were individually functionalized with Poly-L-Lysine (PLL) at a concentration of 15µg/mL (Cat No. 25988-63-0, Sigma-Aldrich, MO) diluted in DPBS. A 100 µL volume of this extracellular substrate was loaded into the devices via syringe and allowed to crosslink at 37 °C overnight in a 5% CO_2_ incubator. Excess PLL solutions within gLL interstitial spaces were then aspirated out and devices cleaned via manual PBS wash using a 1 mL syringe. A rMC-1 solution of 1 × 10^6^ cells/mL was seeded into the sink reservoir and microchannel of gLL devices using a 1 mL syringe and left to adhere for 2–4 h. Solutions of EGF (100 ng/mL, SRP3196, Sigma Aldrich) or VEGF (100 ng/mL, SRP3182, Sigma Aldrich) were added to the gLL source reservoir using a micropipette and allowed to transport overnight along the adjoining microchannel towards the sink reservoir. The concentrations were selected as per previous results from our lab demonstrating the highly chemotactic response of rMC-1 cells towards these fields in comparison to other concentrations of these growth factors [32]. The distribution of ligand within the microchannel was measured to reach steady-state overnight, i.e., exhibit changes in concentration <5%, as previously published by our group [33,42,43,44,45].

### 2.6. Measurement of Cell Migration in the gLL

Cell migration in response to EGF or VEGF stimulus was examined using four testing conditions: (i) rMC-1 stimulated by EGF signaling fields; (ii) rMC-1 stimulated by VEGF signaling fields; (iii) rMC-1 exposed to VEGF for one hour and then stimulated by EGF signaling; (iv) rMC-1 exposed to EGF for one hour then stimulated by VEGF signaling; and (v) control (i.e., media only). Live and adherent rMC-1 were imaged every 30 min for a total of 6 h to record the cell trajectories and total distances traveled in response to applied chemical stimulus. Only rMC-1 cells identified at time t = 0 were tracked for quantification of migration for each group, to avoid recording the path of cells that may have proliferated over the short time-span. We note, however, that our previous work [32,33] indicate rMC-1 proliferation is negligible within our microfluidics system over the experimental time scale used. Cell distances traveled were measured using the conventional parameter of path length, PL, defined by Equation (2),
(2)$$PL=\sum_{i=1}^{n}\left(\sqrt{{\left(x_{i+1}-x_{i}\right)}^{2}+{\left(y_{i+1}-y_{i}\right)}^{2}}\right)$$
where i represents the time point (n = 12), and (x,y) denotes rMC-1 spatial position within the microfluidic gLL system. The sum of displacements between two consecutive points, (x_i_,y_i_) and (x_i+1_, y_i+1_) are then added to determine the total path length or distance travelled by cells during testing.

### 2.7. Computational Modeling

Concentration gradients of VEGF in transwell assays and in the gLL were modeled in COMSOL Multiphysics 5.3a (COMSOL Multiphysics 5.3a, COMSOL Inc., Burlington, USA) using physical properties of VEGF and the true geometry of the assays, as per manufacturer references and our own measurements (Cat No. 3464, Corning Incorporated, NY). Transport of EGF and VEGF within the gLL and transwell assay was computationally modeled to predict reagent concentration over time across the microchannel and permeable membrane, respectively. The diffusivity of EGF was estimated to be 2.0 × 10^−6^ cm^2^/s for EGF [45], and 9.0 × 10^−7^ cm^2^/s for VEGF using the Stokes–Einstein equation [46,47,48].

### 2.8. Imaging and Software

An Inverted transmitted light microscope (Nikon TE2000, Nikon, Melville, USA) was used to observe rMC-1 behavior over time and to perform optical analysis with a cooled CCD camera (CoolSNAP EZ CCD Camera, Photometrics, Tucson, AZ, USA) of a 20× objective magnification (Nikon Plan 20×, Morrell Instrument Company Inc., Melville, NY, USA). A ZEISS LSM 800 confocal microscope at 63× with an oil immerged objective was used for immunocytochemistry (ICC). Brightfield images of rMC-1 were evaluated using ImageJ (NIH imaging software). Immunostained rMC-1 images were assessed using ZEN Blue 2.5 software (ZEISS imaging software) to evaluate the fluorescence intensity.

### 2.9. Statistical Analysis

One-way ANOVA was used to analyze statistical significance among parametric data sets, whereas Kruskal–Wallis rank sum was used for non-parametric ones. Each data set was gathered from a minimum of *n* = 10 to 15 cells per device, using 5–7 independent devices per experimental condition. Values are reported using mean and standard deviation. The post-hoc Tukey and Dunn with Holm correction tests were used to determine statistical significance between conditions, where *p*-values < 0.05 were denoted by an asterisk, *, and *p* < 0.01 were marked with a double asterisk, **.

## 3. Results

### 3.1. Gene Expression via qPCR

The first set of experiments measured the effect of receptor upregulation in rMC-1 upon stimulation with EGF, VEGF, fibroblast growth factor 2 (FGF2), and fibroblast growth factor 8 (FGF8). Values were normalized against controls and shown in Table 2. As seen, rMC-1 exposed to exogenous EGF exhibited increased expression of EGF-R by 2.2-fold. FGF2 and FGF8 stimuli increased rMC-1 expression of EGF-R by a 2.7- and 9.3-fold, respectively. However, EGF-R expression was dramatically increased 18.9-fold in rMC-1 stimulated with VEGF. Additionally, VEGF-R was upregulated 2.8-fold when rMC-1 were stimulated by its cognate VEGF ligand, but downregulated to 0.2 that of basal levels when exposed to EGF. Similarly, FGF-2 and FGF-8 downregulated VEGF-R expression to 0.7 and 0.4, respectively. Lastly, rMC-1 stimulation with VEGF ligand produced an upregulation of FGF2-R and FGF8-R more than the respective cognate ligands. As seen, FGF2-R expression in response to FGF2 stimulus was 1.3-times that of basal conditions, but 5.2-fold higher in response to VEGF stimulus. Similarly, FGF8-R expression was 1.3 times that of basal conditions in response to the FGF8 ligand but 6-fold higher in response to VEGF.

### 3.2. EGF Receptor Expression

We next examined the rMC-1 expression levels of EGF-R in response to both EGF and VEGF stimuli using immunocytochemical staining (ICC). As shown in Figure 3, punctate staining was abundantly distributed over the cell cytoplasm in all rMC-1. rMC-1 stimulated with EGF and VEGF, individually, displayed increased expression levels of EGF-R compared to basal levels (serum-free DMEM). Intensity levels (in arbitrary units, AU) were measured as I_C_ = 0.27 ± 0.098 for control, I_E_ = 0.37 ± 0.063 in response to EGF signaling, and I_V_ = 0.46 ± 0.074 for rMC-1 stimulated with VEGF. Statistical significance was found between the EGF and VEGF-treated groups with respect to control (*p* < 0.01).

### 3.3. Extracellular Signaling Fields

We next used two different technology platforms to create extracellular gradients of the studied ligands in transwell assay (TA) and gLL platforms. Both systems were used because they each create different concentration gradient profiles and have been widely used in chemotactic studies of neural cells [33,49,50]. Concentration gradient profiles (CGPs) for VEGF along transwell assays, VEGF_TA_, and in the gLL microfluidic system, VEGF_gLL_, were modeled over time. We note that CGPs for both ligands were produced, but only VEGF is shown for illustration. Figure 2 shows the VEGF concentration gradient profiles generated within the two different migration assays over the course of 18 and 48 h. The dotted vertical lines in TA of Figure 2B and the gLL graphs of Figure 2D denote the boundaries of the region where the concentration profiles were computed. These regions correspond to the thickness of the transwell assay membrane, Th, and to the length of the microchannel in the gLL device, l. As seen, VEGF concentration across the transwell assays, VEGF_TA_, decreases rapidly to produce very shallow concentration gradients that approached zero (i.e., no gradient). In contrast, the distribution of VEGF within the gLL microfluidic system, VEGF_gLL_, slowly evolved over 48 h to produce quantitative changes in signaling fields. As shown, distinct, non-linear distributions of VEGF were produced along the channel length, l, at each time point from 6 h to 48 h. Further, each gradient profile in the gLL produced both shallow and steep gradients of VEGF at different spatial positions over time.

### 3.4. rMC-1 Chemotactic Responses to Signaling from EGF and VEGF

Our study next examined rMC-1 migration within both TA and gLL microfluidic platforms. Migration was measured in response to EGF signing fields, VEGF signaling fields, and a mixed condition of VEGF-treated rMC-1 responding to EGF signaling fields. Figure 4 displays the number of rMC-1 in TA that migrated towards control, EGF, VEGF, and the mixed condition of rMC-1 pre-treated with VEGF prior to stimulus with EGF fields. Figure 4A shows the average number of motile cells, N, in the conventional TA platform. The average number of rMC-1 that migrated in control conditions (C) was ^TA^N_C_ = 12.1 ± 4.23 µm. rMC-1 migration towards EGF signaling (E) in TA was measured as ^TA^N_E_ = 36 ± 12.8 µm; migration towards VEGF signaling (V) was measured as ^TA^N_V_ = 87.2 ± 25.8 µm. Numbers of motile rMC-1 responding to the mixed VEGF/EGF signaling condition (VE) were measured as ^TA^N_VE_ = 77.1 ± 44.4 µm and rMC-1 stimulated to the mixed EGF/VEGF condition (EV) were measured as ^TA^N_EV_ = 39.7 ± 17.7 µm. Assessment for data normality was performed using the Shapiro–Wilk test, which concluded the data did not fit a normal distribution. Hence, the Kruskal–Wallis rank sum test was used to evaluate statistical significance, and a Dunn post hoc test with Holm correction to discern significance among groups. ^TA^N_V_ and ^TA^N_VE_ demonstrated statistical significance against control (^TA^N_C_). The corresponding *p*-values were found to be 0.0295, 3.206 × 10^−8^, 3.41 × 10^−6^ and 0.0145 for the groups ^TA^N_E_, ^TA^N_V_, ^TA^N_VE_ and ^TA^N_EV_ against control, respectively.

Migration in the gLL system enabled study via average path length, PL, and percentage of motile cells, MC. The average distances traveled in response to signaling produced by the gLL microfluidic system are shown in Figure 4C. Path length, PL, of motile rMC-1 are compared against control conditions (DMEM only) in the device with ^gLL^PL_C_ =14.2 ± 0.5.7 µm. As seen, rMC-1 migrated towards EGF (E) with an increased path length of ^gLL^PL_E_ = 39.7 ± 8.96 µm, and towards VEGF (V) with ^gLL^PL_V_ = 74.7 ± 17.7 µm. Motile rMC-1 in the mixed condition of VEGF/EGF stimuli (VE) produced an average value of ^gLL^PL_VE_ = 56.3 ± 6.64 µm, and the mixed condition of EGF/VEGF stimuli (EV), ^gLL^PL_EV_ = 28.96 ± 5.6 µm. Statistical significance was measured across all groups with respect to control: where ** symbolizes *p* < 0.01 and * for *p* < 0.05. Holm–Bonferroni method yielded *p*-values of 1.79 × 10^−6^, 0, 1.03 × 10^−11^ and 1.88 × 10^−3^ when compared to control for the groups: ^gLL^PL_E_, ^gLL^PL_V,_
^gLL^PL_VE_ and ^gLL^PL_EV_, respectively. Figure 4D denotes the percentage of motile to non-motile cells in the gLL, where motility was defined as movement greater than 20 μm, or 2 cell diameters. The percentage of motile cells in the gLL, MC, was measured as ^gLL^MC_C_ = 46.67% for control (C), ^gLL^MC_E_ = 93.33% for EGF stimulation, ^gLL^MC_V_ = 100% for VEGF stimulation and ^gLL^MC_VE_ = 93.33% for VEGF pre-treatment towards EGF. Standard error was between 3%–6% of the mean values in all cases. Note that MC values cannot be calculated for TA platforms since this test only provides the number of motile cells adhered to the membrane, and not the path length of their individual migration.

## 4. Discussion

### 4.1. rMC-1 Cells as an In Vitro Model of Müller Glia Cells

The experiments performed in this project used cultured cells to model primary MG via the rMC-1 model (Kerafast, ENW001), an immortalized MG cell line derived from adult rats. Although this cell line has been characterized for over a decade, recent research has characterized again their genotype to verify their nature, including T-antigen verification, gene expression and protein expression of molecules including GFAP, CRALBP and vimentin [37,38]. The rMC-1 cell model has been used in diabetic retinopathy models due to its increased expression levels of GFAP and glutamine synthetase, particularly associated with reactive MG [51]. Further, the published literature illustrates that both the levels of VEGF are upregulated and levels of pigment epithelium-derived factor (PDEF) are downregulated in rMC-1 cells, to correlate with the expression levels and patterns in their in vivo counterparts in diabetic retina [52]. Recent studies have also compared the gene expression of more than 9000 genes in rMC-1 cells to demonstrate that approximately 91% of the genes evaluated were not differentially expressed compared to primary MG. That is, a 91% alike genotype was recorded between rMC-1 cells and primary MG [53]. In addition, the use of cultured cell models has been widely accepted for the study of cell behaviors within controlled environments, such as microfluidic devices [32,33]. Use of primary MG has been shown to present difficulties with early senescence and low survival rates [54,55] that diminish cell viability and response to applied stimuli. However, variability in response between primary and cultured cells remains a significant challenge to any modeling of in vivo MG responses to retinal stimuli. The data gathered in this work also aim to help illustrate the use of microfluidic systems to better characterize changes in cellular behavior under constrained conditions, as found in the retinal microenvironment. Our group has previously used the rMC-1 model to characterize their morphology and directed migration within confined environments. Here we demonstrate that rMC-1 cells survive within a diffusion-limited environment of nutrients and growth factors. Future steps will develop more transitional projects by using primary MG.

### 4.2. VEGF-Targeted Therapies in Retina

VEGF is a key angiogenic molecule essential to both reparative and pathological processes in the visual system. While VEGF can promote cell viability and connectivity in retina [56,57], it is well known to accelerate the progression of retinal diseases, such as diabetic retinopathy and wet age-related macular degeneration (AMD), as well as advance neovascularization after injury [58,59]. As a result, many therapies have begun to include anti-VEGF treatments for a variety of retinal disorders with modest, but promising, results. However, the impact of modified levels of VEGF expression on retinal MG cells remains incompletely understood. The effects on MG are particularly significant because, in addition to structural and synaptic support of retinal neurons, MG respond to retinal insult via a set of complex cellular and molecular changes called gliosis. MG gliotic responses can be neuroprotective, creating a physical and biochemical barrier to protect healthy neurons from injury [60], or be neurodegenerative, when chronic response accelerates glial scarring and loss of synaptic function [61]. However, quantitative analyses of mechanistic differences between the two types of responses are needed to elucidate VEGF mechanism(s) of action on MG behaviors [62,63], as the delivery, dosage, and duration of anti-VEGF pharmacology may bypass and/or alter innate, neuroprotective responses of MG.

### 4.3. Relative Receptor Expression

EGFR upregulation in the presence of VEGF has been well studied in glial tumors [64,65], where EMT transition can be initiated for oncogenesis. However, few projects have examined the converse set of reactions where VEGF expression can induce upregulation of EGF-R. The EGF-R signaling pathway is central to both development and pathology of the visual system, and its chemotactic effects on retinal cells during repair have only been recently explored as part of chemotaxis modulation for neuroprotective therapies [32,33]. Quantitative study of the relationship between VEGF and EGF-R signaling in rMC-1 motility, both neuroprotective and neurodegenerative processes, is significant to the development of contemporary therapies focused on anti-VEGF therapies.

Our study first examined gene expression of EGF-R in response to key factors associated with angiogenesis. Dramatic upregulation of EGF-R DNA was measured upon VEGF stimulation than that recorded in response to EGF, FGF2, or FGF8 stimulation via qPCR (Table 1). These data illustrate that VEGF stimulated upregulation of each receptor more than the respective cognate ligand. Such data have been unreported for these molecules in retina and highlight potential cross-talk that requires future study. Immunocytochemistry staining similarly exhibited the largest upregulation of EGF-R when rMC-1 were treated with VEGF rather than its cognate EGF ligand (Figure 3). These results support an earlier published study that showed angiogenic factors are able to induce higher EGF-R expression through shared pathways [66]. Recent reports have also established the relationship between downstream extracellular signal-regulated kinase 1 and 2 (ERK1/2) signaling on elevated VEGF concentration in MG in diabetic retina [67,68].

### 4.4. Müller Glia Migration Ability within the In Vivo Retina

MG cells in mammals respond to injury by proliferating, extending their processes beyond the outer limiting membrane, and migrating in later stages of retinal degeneration [69,70]. Breaking down of the blood retinal barrier creates a concentration imbalance of growth factors such as VEGF and neurotransmitters such as glutamate, which in higher concentrations become neurotoxic. MG readily respond to changes in the retinal homeostasis through gliosis, characterized by cellular hypertrophy and upregulation of GFAP. Neurotoxicity within the retina rapidly degenerates photoreceptors, which are phagocytosed by MG during the first hours of damage, while microglia migrate towards the affected area to overtake phagocytosis [71]. In addition, upregulation of glutamate has been demonstrated to induce MG to re-enter the cell cycle by upregulating Cyclin D1 and Cyclin D3 and proceed to migrate from the inner nuclear layer (INL) towards the damaged area [72]. Injury models of the retina have quantified the migration of MG in vivo over several days, post-injury, to demonstrate the ability of MG to translocate their center of mass from the inner nuclear layer (INL) to the outer nuclear layer (ONL) [73]. Full translocation of MG, where the cells detach from either side of the inner or outer limiting membrane, has not been evaluated. Most injury models have characterized the reaction of MG upon damage to the ONL and their response to changes in the microenvironment’s homeostasis and photoreceptor death. Nevertheless, research suggests that reduction of stiffness in the limiting membrane by conditions like macular pucker would weaken the adhesion forces of MG, affecting their mechanotransduction [74], which would facilitate complete cellular migration in the direction of the stimulant’s highest concentration gradient. Additional study is needed to quantitatively and mechanistically examine the fine mechano-sensing abilities of MG in response to retinal insult and injury.

### 4.5. Exogenous Signaling Fields of VEGF

Migration of rMC-1 in response to VEGF and EGF signaling was next evaluated using both transwell assays and our microfluidic assay, the gLL. TA, such as Boyden chambers, have been well utilized in chemotactic study to evaluate the chemotactic strength of different ligands and/or ligand combinations upon a variety of cell types [75,76]. However, TAs operate via passive diffusion through a permeable membrane, where cells migrate through pores and attach upon the membrane underside. Further, TAs generate highly complex gradient fields that are difficult to control and/or model quantitatively over time. Previous work [77] has shown that concentration gradients are poorly retained in this platform after ~ 6 h, as diffusive transport of molecules reaches equilibrium across the thin membrane quickly to produce very shallow signaling gradients. Our analysis similarly exhibited the rapid decline of signaling gradients in the TA platform when developing concentration gradient profiles (CGP) of high molecular weight growth factors such as VEGF [78,79]. By contrast, our work used the microfluidic gLL to produce well-defined CGP for over 18 h in comparison to the transwell assay. Microfluidics enable precise spatial and temporal control of factors that drive migratory processes and are particularly advantageous for modeling changing retinal gradients developed over small anatomical distances [80,81]. Difference in CGPs enable the study of rMC-1 response to various rates of change in concentration and gradients, important in the modeling of therapeutic and pathological transport within retinal microenvironments. Molecular transport is particularly significant in models of diabetic retinopathy and wet AMD, where molecules and growth factors diffuse at different rates and are highly correlated with VEGF concentration and temporal expression [82,83]. In addition, the gLL is able to model exogenous stimulation via diffusion, convection, and/or advection, used for delivery of pharmacology within the constricted retinal environment. The tunable gLL also enables the study of transient concentration changes, which can be retained for long periods of time to facilitate analysis of cellular behavior and response over time.

### 4.6. VEGF-Augmented Chemotactic Responses to EGF Signaling

In both platforms, VEGF induced much greater numbers, N, of motile rMC-1, as well as greater fractions of motile cells, MC, and larger migration distances, PL, as measured by the gLL. However, rMC-1 pre-treated with VEGF exhibited significantly higher N in the TA platform, as well as higher PL along EGF fields in the gLL device (Figure 4). This corroborating data elucidates crosstalk between VEGF and EGF-R signaling pathways to regulate movement. This is significant because previous studies have reported elevated levels of endogenous-derived VEGF in MG with diabetic retinopathy alter gliosis responses [21,84]. Our data highlight the need for detailed investigation of the molecular interaction between VEGF and EGF-R to improve contemporary treatments and advance the development of regenerative therapies.

Taken together, our study used quantitative platforms such as the gLL and transwell assays to point out the interplay between VEGF stimulation and subsequent EGF-R upregulation in rMC-1 chemotaxis. Results illustrate that treatment of VEGF in rMC-1 led to chemotactic responses, which could help to gain insight on the role of VEGF and the consequent reactive behavior in MG in DR. Further exploring signaling pathways that link EGF-R upregulation with elevated concentrations of VEGF will aid current development of retinal therapies.

## Figures and Tables

**Figure 1 brainsci-10-00330-f001:**
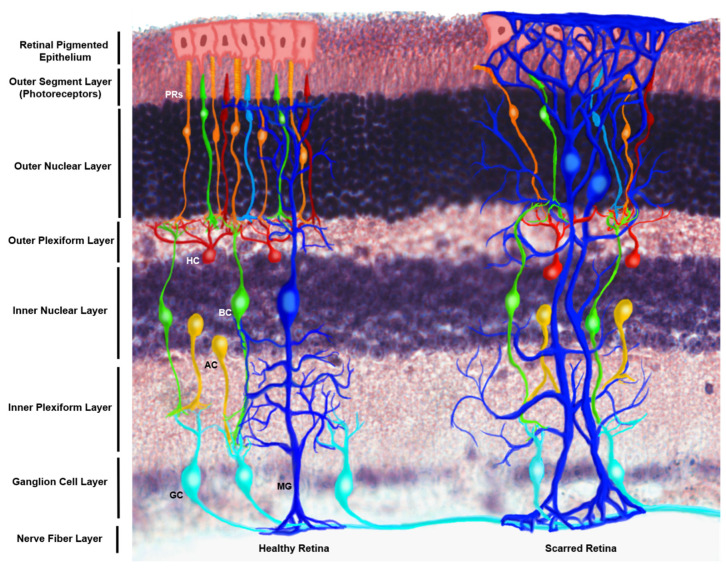
Schematic of the structural distribution of Müller glia (MG) within the healthy and scarred mammalian retina. This figure makes use of an immunostained retinal cross section of the N1CR2 murine model, created to drive up green fluorescent protein (GFP) in interneuron progenitors without changes in retinal structure, function or intrinsic cellular expression based on the transgene [10]. From the posterior to anterior side of the retina (bottom to top) end-feet of MG can be appreciated at the nerve fiber layer, extending their processes through the inner plexiform layer to the inner nuclear layer, where their nuclei reside. MG processes extend up to the interception of the outer nuclear layer and the outer segment layer, constituting the outer limiting membrane, which serves as a biochemical and physical barrier for the retinal tissue. (Left) Healthy retina featuring the nerve fiber layer, which is made of ganglion cells (GC) axons, whose bodies reside in the ganglion cell layer, followed by the inner plexiform layer where ganglion cells connect with amacrine cells (AC) and bipolar cells (BC), whose cell bodies are found in the inner nuclear layer; followed by their connection to horizontal cells (HC) and photoreceptors (PRs) at the outer plexiform layer. Bodies of photoreceptors align in the outer nuclear layer, while the photoreceptor disks lie on the outer segment layer, adjacent to the retinal pigmented epithelium (RPE), which selectively permeates molecules from the choriocapillary network. (Right) Scarring of the anterior side of the retina is characterized by the extension of MG processes beyond the outer limiting membrane into the RPE, which combined with extracellular matrix deposition and addition of glycoproteins comprise the scar that displaces the endogenous cells of the tissue and impedes regeneration.

**Figure 2 brainsci-10-00330-f002:**
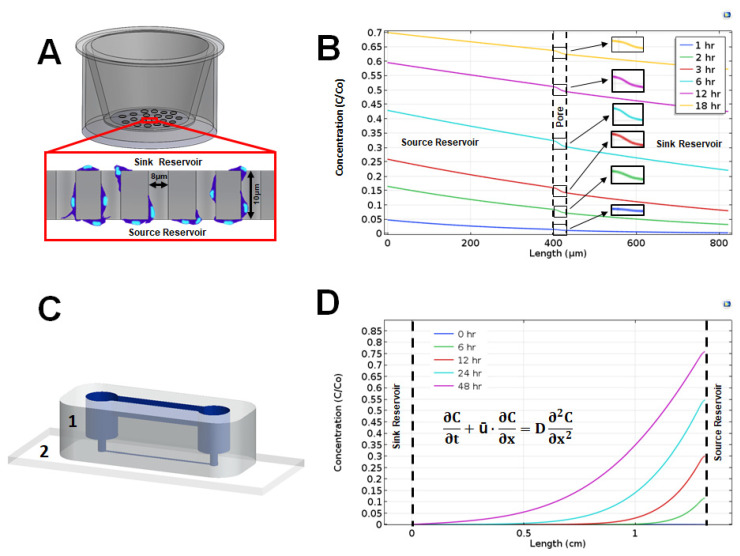
Distribution of vascular endothelial growth factor (VEGF) ligand along the characteristic length of transwell assays and the glial line system (gLL) microfluidic system over time. (**A**) Schematic of a transwell assay with the porous membrane shown in the inset. Representative cells (in blue) migrate from the top sink reservoir to the bottom source reservoir through the pores of the membrane that divides the two. Motile cells are imaged on the underside of the membrane. (**B**) Concentration profile of VEGF across the transwell membrane at 1, 6, 3, 6, 12, and 18 h, normalized to inlet concentration, Co. (**C**) Schematic of the gLL device constructed using elastomeric polydimethylsiloxane (PDMS) (1) bonded upon chemically cleaned glass coverslips (2). The structure of the microfluidic system is highlighted in blue. (**D**) Normalized concentration profile of VEGF across the microchannel of the gLL at 0, 6, 12, 24, and 48 h. The concentration profile is defined by transport processes modeled via the convective-diffusion equation shown.

**Figure 3 brainsci-10-00330-f003:**
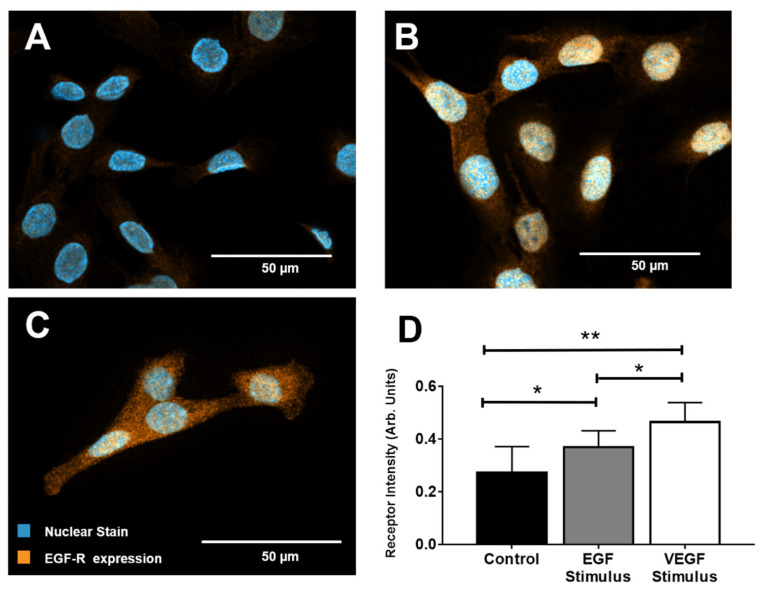
Epidermal growth factor receptor (EGF-R) expression in rMC-1 upon stimulation with epidermal growth factor (EGF) and vascular endothelial growth factor (VEGF). (**A**) Representative image of EGF-R expression in rMC-1 at basal conditions. (**B**) Image of EGF-R within rMC-1 after stimulus with EGF for 1 h. (**C**) Image of EGF-R expression in rMC-1 following VEGF stimulus for 1 h. Orange denotes EGF-R molecules while blue marks nuclear staining (DAPI). (**D**) EGF-R expression measured using fluorescence intensity in arbitrary units (AU). A minimum of 15 cells per condition were used for these calculations.

**Figure 4 brainsci-10-00330-f004:**
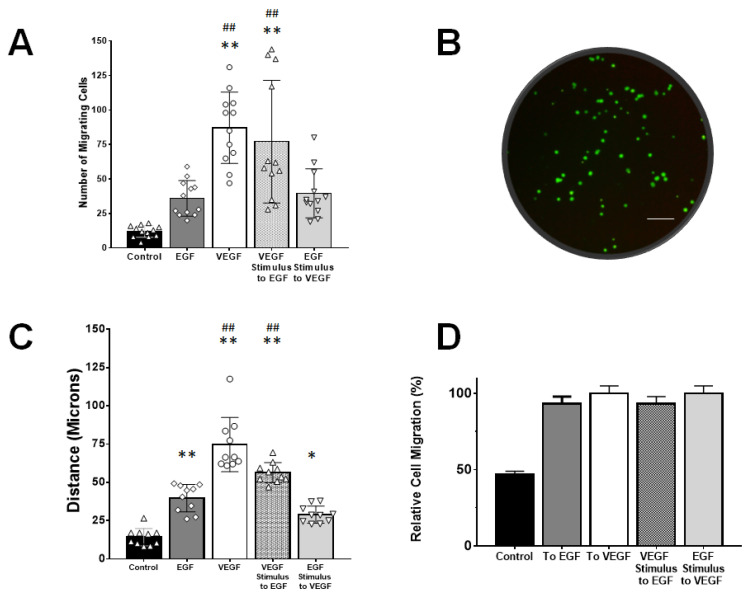
Müller glia migration in response to exogenous signaling produced within transwell assays (TA) and the gLL microfluidic system. (**A**) Fold-change in numbers of rMC-1 that migrated towards concentration gradient profiles of VEGF and EGF established with TA. Numbers of motile rMC-1 were also measured in response to control (no gradients, media only) and a mixed signaling condition of rMC-1 pre-treated with VEGF responding to EGF signaling fields. (**B**) Bottom of TA membrane displaying fluorescently labeled cells that migrated through it. Scale bar is 200 µm. (**C**) Average path lengths, or accumulated distances, of rMC-1 that became motile in response to signaling fields of EGF and VEGF created within the gLL microfluidic system. Conditions of mixed VEGF/EGF signaling and control were also produced as with the TA platform. (**D**) Average fraction of motile cells in response to signaling fields within the gLL system. Statistical significance denoted by * (*p* < 0.05) and ** (*p* < 0.01) with respect to control; ## represents statistical difference (*p* < 0.01) among the groups with respect to the EGF group.

**Table 1 brainsci-10-00330-t001:** Gene regulation examined via quantitative polymerase chain reaction (qPCR).

Gene	Primer Sequence (5′–3′)	M.W.(bp)
**FGF2**	**F:** GAACCGGTACCTGGCTATGA	**182**
	**R:** CCGTTTTGGATCCGAGTTTA	
**NRP-1**	**F:** GCTACCCTCATTCTTACCATCC	**119**
	**R:** GCAGTCTCTGTCCTCCAAATC	
**FGFR3**	**F:** CTGTATGTGCTGGTGGAGTATG	**98**
	**R:** CTGCAGGCATCAAAGGAGTAA	
**EGFR**	**F:** GCTGTGCGATTTAGCAACAA	**146**
	**R:** GGACAGCTCGGATCACATTT	
**Nestin**	**F:** CAGTACTCGGAATGCAGCAA	**98**
	**R:** CTTCTGTGTCCAGACCACTTT	
**GFAP**	**F:** CACCCTGCATCTCCAACTAAC	**109**
	**R:** GGAAGAAAGAGGAAAGACAGGG	
**GAPDH**	**F:** ACTCCCATTCTTCCACCTTTG	**105**
	**R:** CCCTGTTGCTGTAGCCATATT	

A listing of the genes encoding molecules and interfibrillary proteins studied, alongside primer sequence (5′ to 3′) and size in base pairs (bp). All tests were measured using glyceraldehyde 3-phosphate dehydrogenase (GAPDH) as the standard.

**Table 2 brainsci-10-00330-t002:** Gene expression of cognate receptors in rMC-1 stimulated by a panel of chemotactic ligands.

	EGF-R	FGFR-2	FGFR-8	VEGF-R
EGF	**2.2**	2.5	0.7	0.2
FGF2	**2.7**	1.3	0.9	0.7
FGF8	**9.3**	2	1.3	0.4
VEGF	**18.9**	5.2	6	2.8

Receptor expression rMC-1 stimulation with epidermal growth factor (EGF), fibroblast growth factor 2 (FGF2), fibroblast growth factor 8 (FGF8) and vascular endothelial growth factor (VEGF) ligands evaluated via quantitative polymerase chain reaction (qPCR). The bold values highlight epidermal growth factor receptor (EGF-R) expression following 1 h stimulus with selected ligands. All data are normalized with respect to basal conditions (control).

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
