# Peer review of "VEGF Upregulates EGFR Expression to Stimulate Chemotactic Behaviors in the rMC-1 Model of Müller Glia"

_brainsci, 2020, doi:10.3390/brainsci10060330_

Round 1

Reviewer 1 Report

The authors have substantially improved the manuscript. However, I still have serious concerns about using an immortalized cell line and drawing any conclusions from it without comparing the analyses to real Muller glia cells. For some reason, it seems that the authors are very resistant to try experiments using MG primary cultures. 

I think that given this, the paper could potentially be accepted but to avoid overstatements, the cells shouldn't be called MG but rMC-1 or at least MG-like. This includes the whole paper and the title.

Reviewer 2 Report

VEGF upregulates EGFR expression to stimulate chemotactic behaviors in Muller glia.

The authors did a very thorough and extensive revision. Most points were addressed. One major request was not done but is still necessary and moreover, now I am not sure where this is supposed to go, based in the new paragraph in the discussion. I think I understand what the authors want to say but it is actually self-defeating. There are also a few other little points:

Line 150 – 2.4 Methods

Where does the N1CR2 murine model suddenly comes from? Why this mouse? This was only used to crease Figure 1? This is really confusing because this implied in vivo data but this procedure did not create data. Besides, a normal C56BL6 would have been the better choice – now questions come up such as “Is that mouse resembling a true wild type?” The effort is very  praisable though! Adding a full methods section just because of an image for illustration purposes is not necessary and again just leads to confusion. Method sections describe methods used to generate data. For illustration purposes, the info can be added in the legend of Fig.1 as it’s done in review articles (that don’t have a methods section). For example: Background shows a cross section of an adult(?) mouse retina stained with H&E stain to indicate retinal layers and overall structure. Sample was provided by .. .

The legend says human retina – but it’s mouse?!  (A human retina has several layers of RGC – this only has one!)

Please add healthy and scarred in the pictures  – define scarred.

Line 122 – I am sure it’s the other way around! Trizol treated MG won’t migrate.

Line 367 – paragraph 4.1 discussion

 The criticism was that an immortalized cell line was used, that by definition has high proliferation rates. No characterization was performed – as requested. If cells, animals whatsoever biological samples are used as major tool, and here is THE tool, this has to be characterized in this study. Citations are not acceptable, especially not for proofing characteristics for cell cultures, which always differ from badge to badge, culture medium, time of culture process (and many more factors) and especially not with the references given here – but this will be addressed later. The Sarthy paper is from 1998 and they only characterized with GFAP and Cralbp – that’s 20 years ago. Just using a cell line but not confirming original cell characteristics is a major criticism that made made before and still remains.

If the authors claim to have MG, their identity has to be proven – in this manuscript.

  • IHC is used for this manuscript with other markers – please confirm GS, GFAP and Cralbp expression of your MG for the time points of experiments performed – the markers shown in the cited other papers. Show co-localization with EGF – Fig. 3.
  • Since sometime transcript and protein levels are not conform, please also show qPCR data can confirm no alterations in glial genes when migrating – which is important as well for this study.

In the discussion paragraph, the authors say now, that primary cultures are difficult because they have “.. slower proliferation rates..”, which is actually the characteristic they show in vivo. MG do not proliferate in mammals. But then, proliferation is not required for this experiment – which is about chemotaxis - and also, only a few cells will be put in the device therefore confluent cells are not necessary. Primary MG do live for several weeks in vitro – even without proliferation, they don’t die fast. Therefore, all these arguments are actually self-defeating and the use a more artificial system (immortalized cell culture) will not help to reveal MG behavior in vivo in mammals incl. humans at all. So what is the point? With this argumentation, the authors are actually moving away from the their overall goal, understanding gliosis in vivo. If the main argument is that the more artificial system is preferred because it “behaves” better than the more natural model. How will that ever be transferred to clinical approaches? This needs to be rewitten.

First please characterize the cells yourself and show that they are what they are supposed to be in your assay and under your conditions. This is a requirement for every study – no matter where else cited before. Then use the argument found in the answer to Reviewer 1 about the “proof of principal” using immortalized cells first – saying stablished and used in other biomedial assays (own or others) before transferring the knowledge to primary cultures that need to be obtained from animals -  but will be done next. This is legit to say.

With regard to the cited papers – these are not good choices. The citation that compared primary cultures to immortal cells is useless. First, in this paper has NO DATA at all - only procedure is described and links to data referring to a different paper. There is not one sentence saying anything about 9000 genes and also not the nature of these genes, not one glial gene is named. The biggest problem with that paper however is, it weren’t primary MG cells used – the cells were harvested from P2-4 rat eyes, a time point where no glia is present – at least no MG. These were retinal progenitor cells that where probably maintained.

It is known that MG and RPC share about 80% of their gene expression profile (Roesch 2008, Jadhav 2009 from the Cepko Lab). Growing RPCs and passaging them will not result in glia that resemble an in vivo glia cell. So no idea what they cells they profiled and sequenced but if that shows 91% overlap to immortalized MG, it’s a proliferating cell type – and no mature MG. Also two markers characterization is not truly not well-characterized. I would agree with 4-5 markers and different assays qPCR and WB as the simpler ones and RNA-seq as the more sophisticated method, if performed. The Thompsen paper only reports a GS WB band – that is one marker and one technique – and poor characterization. Please revise this paragraph and add the missing data.

Fig3 – label still too tiny and not readable – have at least 7 point font.

The vast majority of cited papers in this manuscript contradict the results from labs that have reproduced their data and published in renowned journals – which are missing here. The specific conditions – treatment – injury – what injury and more importantly - what species is not clarified and all is mixed up. This specification however is essential when talking about proliferation assays and migration of MG – but then again, proliferation is not even the topic of this manuscript and should be left out completely.

Author Response

This manuscript is a resubmission of an earlier submission. The following is a list of the peer review reports and author responses from that submission.

Round 1

Reviewer 1 Report

In the present manuscript, Peña and Vazquez identify a novel crosstalk between VEGF and EGFR as a treatment with VEGF increases the expression of EGFR by qPCR and by immunostaining. While this finding may be potentially interesting, there are many fundamental weaknesses in the paper in its present form:

1- An immortalized cell line is not a good representation of the physiological behaviors of the Müller glia, and the title  "VEGF upregulates EGFR expression to stimulate chemotactic behaviors in Müller glia" is misleading as these cells are not Müller glia but the rMC-1 cell line. In order to make the paper meaningful, the authors should repeat the qPCR analyses in primary Müller glia cultures and the EGFR immunos in explant cultures or intact retinas.

2- The chemotactic behaviors descrived are probably an artifact due to the utilization of a cell line or the culturing conditions as MG within the retinal tissue only migrate radially in some conditions (e.g. photoreceptor degeneration), they have very different morphologies from the ones shown in the pictures, and are attached to the OLM and ILM.

Also, more information is needed on the experimental paradigm for the chemotaxis experiment: how many cells were analyzed? how many experiments?

3- EGF has been shown to stimulate MG proliferation (e.g. Ueki, Glia 2013). I am wondering if this effect has been taken into consideration for the chemotactic experiments. 

4- Figure 1 has nothing to do with the paper and I am puzzled at why the authors included it. In the diagram, the ONL is very thin compared to the inner retina and there are typos (it is Bruch's membrane and not Brush's membrane). 

A schematic of the layers of the retina can be found elsewhere and Fig.1 does not add anything to the paper. 

Reviewer 2 Report

Review

Summary: The manuscript “VEGF upregulates EGFR expression to stimulate chemotactic behaviors in Muller glia” from Pena and Vazquez reports about the migration behavior of cultured MG towards the growth factors EGF, VEGF and both in combination. They showed that MG displayed a strong affinity for migrating towards VEGF gradients. This migration is accompanied by up-regulation of EGF-receptor, not only after EGF exposure (ligand) but also after VEGF exposure and that this suggests an interplay that might have relevance for understanding injury mechanisms (gliosis) occurring in a variety of retinal diseases that are caused by dysfunction of the vasculature.

Broad comments: The manuscript is overall a very interesting study since it shows how MG adapt to environmental changes, in this case to growth factors and specifically to VEGF. An immortal cell line of rat MG is used to study these effects. To monitor and measure migration a very interesting model of transwell assays were used, transcription levels are qualified. The manuscript is very detailed and most methods are excellent, however, some parts of the manuscript are somewhat confusing or incomplete and need to be addressed – they are listed below.

Specific comments:

Abstract: Two thirds are background which is too long. A very important section, the methods are not described at all, stainings are only qualitative, not quantitative. Sentences are too long and partly confusing. Please reformulate the sentence from line16-19. Add “the” before “retina”

Introduction: is very detailed and guides very well to the topic. One sentence is confusing – line 55: “Progressive retinal dysfunction often leads to breakdown of the BRB..” it is the other way around, cellular dysfunction leads to BRB breakdown and retinal dysfunction (overall tissue)

Line 81-83 describes the scope of the current project and should not have any citations

Line 83-86 multi-clause sentence that is not clear. In general, please avoid complicated sentence structures. That makes the manuscript partly very hard to read. Also there are too many abbreviations. If no more limit, use full terms throughout your manuscript – it is really difficult to remember all these specific names, and makes it hard to read the manuscript, in particular for transwell assay. gLL is very abstract as well – “glia-line microfluidic system (gLL)” would be good to keep as term throughout the manuscript, especially since that is not a broadly used abbreviation as MG is.

Please clearly state what was done and what is new in more than one sentence. Where does the idea originated to test VEGF and EFG-R?

Lin 74: in “the” retina

Figure 1 – although this schematic has some beautiful art characteristics, the illustration is not anatomically correct. This is a very abstract image and not suitable for a scientific article if not displaying correct dimensions and cell morphologies. The ONL has vessels not illustrated in the image. The morphology of the MG is not correct, they do not have bubble-like protrusions in the GCL, the ONL and INL is way too small, the INL and GCL as monolayer too large. The species should be named– is that mouse or human? If human, cell composition is not correct. It is not obvious at first glance, that disease and normal condition are displayed - healthy should be first and separated from diseased – basically two images. Legend: line 52 “rods” not “Rods, line 51 missing space 

Methods: partly very detailed and exact, the illustration for the transwell assay is very helpful. Ideally the methods are in order – was qPCR and ICC not performed after the migration assay?

Cell line, the fact that rMC-1 is an immortalized cell line needs to be added and this fact, with regard to transfer your results to in vivo disease treatments, needs to be discussed. It is important to have at least additional information of normal primary glia, originating from normal mouse tissue. Although still a culture system, primary mouse cells will be closer to an in vivo MG with regard to gene expression and behavior than an immortalized (mutated) MG line. If that is not possible, overstatements needs to be avoided and pro and contra of the model system explained. All models have downsides but it’s important to be aware of that and make the reader aware of that. Please also note that cultured MG (also primary MG) migrate while MG in vivo do not, at least not in mammals. Also, normal healthy MG on non-albino retinas do not express GFAP. GFAP is an astrocyte marker and only expressed when MG get reactive. This happens when primary MG are plated but after several days of culturing, this marker goes down – in primary MG. If GFAP stays on in this line, this might be an indication that these cells remain reactive and should be discussed, since that is an important fact. Show both reported MG markers for these cells – Cralbp and GFAP to confirm cell identity. For the culture paradigms, please state how you maintain the line what cells you consider healthy (number of passages). Immortal cells degenerate around passage #30 and will not display “normal” cell characteristics anymore. Add pictures of your cultures – wide field (10x) and zoomed in (20x or 40x) phase or fluorescent maker with phase. This can be included in Fig3.

-qPCR: State how many technical and biological replicates were used. Against which house keeping genes was normalized?

ICC – is that immunofluorescence since the images in Figure 3 display fluorescent images. If so, that is the conjugation of the secondary antibody? A MG marker should be used as reference – such as Cralbp, since that marker is present in these immortal MG to confirm cell identity. Please add that DAPI is the nuclear staining line 136.

4 and 2.5 – are these two different assays as implied by separating then or one technique? That is one of the most confusing points in this manuscript. Show pictures of the CellTrackerTM green for your 4 conditions in results – Fig 4

Fig 2 – labeling too small to read – should be at least 8pt final formatting, please add full name of PDMS in legend at the end.

Lines 184+185: how was this concentration determined, were other concentrations tested, if not, why? A dose curve needs to be shown and various concentrations (either tested in the own lab or shown by other groups) should be discussed. Since this is a gradient-based migration, how are the gradient kept stable in the medium? How long is the migration experiment in total? According to Fig 2. 18h? Time points and full time span need to be added in the methods section as well as a brief explanation for, why this paradigm.

7 was EGF first + VEGF second also tested, if not why?

10: excellent n and group sizes but do they apply for all experiments? Concrete numbers for all individual experiments need to be given – qPCR, staining, migration assay. Since multiple comparisons are performed, a correction such as the Holm-Bonferroni correction for multiple testing has to be performed, since the error increases with every test. The p’s need to be revised where necessary after the correction.

Results: Table 1 – the raw data of all genes including house keeping genes, the calculated delta cT and the delta delta cTs needs to be given in supplement to get an idea about the nature of gene expression levels. Glial genes should be included to have a reference of what are high and what are low levels.

One major criticism is that a labeling was used for quantitative analysis. Antibody-body based labelings, especially the indirect technique, are a qualitative method to show where a protein is expressed (location). Quantifications are only possible with a reference but even then questionable. Western Blots or ELISAs are the only reliable protein quantification techniques accepted in the field, as well as Mass Spec. If a semi-quantitative analysis using IHC is used, a glial marker or any protein known to stay consistent (baseline), has to be used as reference and for normalization. Cells needs to be stained with markers for the protein of interest and a reference marker. Since the staining quality and therefore fluorescence intensity depends on fixation, durations of incubations, temperature und many more factors that can vary from experiment to experiment, comparisons between several experiments are problematic. Which is why it is not accepted as quantification method. Moreover, all images( of both markers and all samples at all times) have to be taken with the same paradigms and they need to be stated in the manuscript: exposure time, gain levels, pixels etc. But again, even by doing so, this is not a valid protein quantification technique. This data needs to be revised. A glial marker needs be added and displayed anyway, since the pictures need to show and confirm MG identity.  – for the display - the Figure labeling is not readable – not in the images or the graph. Also control and EGF can’t be significant with that SD as shown – if that is the SD. Indicate whether mean +/- SD or SEM is displayed – for all graphs in all legends.

Figure 4 – add pictures with the green trace to show what was measured – ideally with a distance mark. Revise p’s after correction for multiple testing. Add mean +/- .. . Why is no absolute data shown? If fold change is chosen to display in graphs, add all (raw) data to calculated values in the supplement. I don’t understand graph c, regardless the growth factor, MG migrate similarly? Why are there no error bars in C - line 308-310 says differently.

Line 310 “MC values cannot be calculated for TA platforms” – is that not the platform used? avoid abbreviations if no word limits.

Discussion: there is an error: lines 332 – 334 “ MG gliotic responses.. to aid retinal remodeling and regeneration,.. “retinal remodeling is a universal finding and subsequent to retinal degenerative diseases” see Jones et al., 2013 and a really bad thing, see papers from Robert Marc and Bryan Jones. Moreover, MG in mammals DO NOT REGENERATE, only in fish. MG in mammals need to be reprogrammed in order to get partial regeneration but that requires artificial interventions and /or is restricted too very young ages but still requires environmental manipulations. There is no regeneration in mammals after injury as reported in many papers and reviews. Also MG do not migrate in mammals in vivo. In fact, if they get reactive, they upregulate GFAP and other intermediate filaments which makes them more stiff. Therefore, this migration assay on immortal cells is hardly transferrable to in vivo conditions if there are no other alterations (overexpression of Ascl1 or alterations in epigenetics). However, this does not change the fact that MG respond to VEGF by up-regulating EGF-R and there is no doubt that this might be a very crucial process for gliosis. However, without any tests of MG within tissue (that can be explant cultures or in vivo) no claims can be made, only speculations, based on an immortalized MG cell line. The discussion needs to be revised with regard to true facts obtained from this study with detailed discussion of the limitations of the MG-cell line. Overstatements need to be avoided.

Line 346 – these papers do not report regenerative medicine. Again, there is no regeneration in mammals after injury in a normal not genetically altered mouse or with factors treated. The papers that report reprogramming of MG are not cited at all.

Line 351 ICC is no accepted method for protein expression quantification only WE or ELISA

Lines 390-391: EGF is a mitogenic factor and does not induce neuronal gene expression – it is used to induce proliferation in MG, for instance to generate primary cultures (that also needs to be reprogrammed into neurons using reprogramming factors such as Ascl1)

Please cite correctly: these papers are about YAP nor EGF-R! There is a connection to EGF-R and VEGF which is the topic of this paper. YAP is the factor inducing all events seen in these studies, not EGF-R nor VEGF. Even if YAP bind to EGR-R – if YAP expression is not manipulated, EGF-R has no meaning.

Line 391 – ref 29: Hamon et al 2019: “Contrasting with fish or amphibian, retinal regeneration from Muller glia is largely limited in mammals.  … In the mouse retina, where Muller cells do not spontaneously proliferate, YAP overactivation is sufficient to induce their reprogramming into highly proliferative cells...” – no neurons and genetic alteration of YAP is required; ref 71 Rueda et al 2019 “In response to retinal damage, the Muller glial cells (MGs) of the zebrafish retina have the ability to undergo a cellular reprogramming event in which they enter the cell cycle and divide asymmetrically, thereby producing multipotent retinal progenitors capable of regenerating lost retinal neurons. However, mammalian MGs do not exhibit such a proliferative and regenerative ability. Here, we identify Hippo pathway-mediated repression of the transcription cofactor YAP as a core regulatory mechanism that normally blocks mammalian MG proliferation and cellular reprogramming. MG-specific deletion of Hippo pathway components Lats1 and Lats2, as well as transgenic expression of a Hippo non-responsive form of YAP (YAP5SA), resulted in dramatic Cyclin D1 upregulation, loss of adult MG identity, and attainment of a highly proliferative, progenitor-like cellular state. Our results reveal that mammalian MGs may have latent regenerative capacity that can be stimulated by repressing Hippo signaling.”

Sentence line 398 – 402 makes no sense.

Avoid multi-clause sentences and try to find some synonyms for “illustrate” and “aid” now and then

At this point, with having no evidence that normal primary MG or even MG within the tissue (that can be explants or in vivo) would display the same behavior as artificial immortal MG – all is just speculation and the wording needs to be changed. The discussion needs an extensive revision.

All parts with regard to regeneration should be removed since the data presented does not indicate any relation to a possible regeneration. The connection to disease and vasculature malfunctions – neurodegeneration and (true) retinal remodeling however, is very interesting and should be expended with supporting literature. It would be very beneficial to emphasize this and do more background research about VEGF levels during retinal disease.

Major points that need to be addressed

MG identity Reliable and accepted protein quantification assay Relevant facts and correct classification of this study extensive thorough revision